# Monocytes/Macrophages and Atherogenesis

**DOI:** 10.3390/ijms262210962

**Published:** 2025-11-12

**Authors:** Sergey Kozlov, Tatiana Riazantseva, Ivan Melnikov, Sergey Okhota, Viktoriia Vasenkova, Olga Saburova, Yuliya Avtaeva, Konstantin Guria, Lyudmila Prokofieva, Zufar Gabbasov

**Affiliations:** 1National Medical Research Centre of Cardiology Named After Academician E.I. Chazov of the Ministry of Health of the Russian Federation, 15A 3-rd Cherepkovskaya Street, Moscow 121552, Russia; tasya-blokhina@yandex.ru (T.R.); ivsgml@gmail.com (I.M.); ae2007@mail.ru (S.O.); vasenkova.vika@yandex.ru (V.V.); saburovaos@mail.ru (O.S.); julia_94fs@mail.ru (Y.A.); kgguria@gmail.com (K.G.); prokofieva_l@mail.ru (L.P.); zufargabbasov@yandex.ru (Z.G.); 2State Research Center of the Russian Federation-Institute of Biomedical Problems of Russian Academy of Sciences, 76A Khoroshevskoye Shosse, Moscow 123007, Russia

**Keywords:** monocytes, macrophages, atherogenesis, inflammation, fibrosis, innate immunity, phagocytosis, cell death, tissue repair

## Abstract

Atherosclerosis is a widespread cardiovascular disease characterized by retention of atherogenic lipoproteins in the arterial wall and the onset of subclinical vascular inflammation; the development of atherosclerotic plaques; eventual narrowing of the arterial lumen and/or plaque disruption; and subsequent manifestation with stable ischemia or acute atherothrombotic events. Numerous cell types are implicated in atherogenesis. Monocytes/macrophages are considered pivotal participants in this complex process. They play a crucial role in the onset and augmentation of inflammation and greatly contribute to atherosclerotic plaque growth and destabilization. However, monocytes/macrophages are also essential for the resolution of inflammation and the stabilization of atherosclerotic lesions. In this regard, studies of the function of monocytes/macrophages in relation to this disease are of considerable interest to researchers, as the results can help to design new drugs aimed at preventing the development of atherosclerosis and its complications. This review presents current data on the classification and functions of monocytes/macrophages; discusses current hypotheses regarding the involvement of monocytes/macrophages in atherogenesis; and highlights existing gaps in evidence. This review is primarily aimed at readers with a background in clinical medicine who are interested in the involvement of monocytes/macrophages in atherogenesis.

## 1. Introduction

Atherosclerosis is a widespread cardiovascular disease characterized by retention of atherogenic lipoproteins in the arterial wall and the onset of subclinical vascular inflammation; the development of atherosclerotic plaques; eventual narrowing of the arterial lumen and/or plaque disruption; and subsequent manifestation with stable ischemia or acute atherothrombotic events (Figure 1) [1]. Atherosclerotic cardiovascular diseases remain a leading cause of morbidity and mortality worldwide. Although conventionally regarded as a disease affecting the elderly, recent years have seen a rise in premature development of symptomatic atherosclerosis [2,3]. Current approaches to preventing and treating atherosclerotic cardiovascular disease are based on assessing and modifying traditional cardiovascular risk factors, with a primary emphasis on lowering cholesterol-rich apolipoprotein B (apoB)-containing lipoprotein levels in the blood [1].

Lipid infiltration, modification, and retention in the arterial intima are regarded as key pathogenic processes in atherogenesis. They are tightly interwoven with another essential component of atherogenesis: inflammation. Atherosclerosis is characterized by a low-grade chronic inflammatory response, which is crucial for atherosclerotic cardiovascular disease development. Large randomized clinical trials, such as CANTOS, COLCOT, and LoDoCo2, demonstrated that the suppression of certain inflammatory pathways produces a pronounced decrease in major acute cardiovascular event (MACE) rates [4]. The inflammatory response in atherosclerosis is driven by a number of cell types, of which monocytes/macrophages are pivotal.

Monocytes/macrophages are involved in atherogenesis from its earliest stages [5]. They accumulate in atherosclerotic lesion sites, engulf lipids, and form the hallmark foam cells; are largely involved in cell and tissue destruction and regeneration; and are integral to atherosclerotic plaque disruption or stabilization [6].

In preparation for this paper, PubMed, Medline, and Web of Science databases were searched for publications. The search encompassed animal and human studies, ranging from basic research to randomized clinical trials, meta-analyses, and reviews. Original research articles with consistent methodology published from 2010 to the present were given priority, although some earlier articles providing crucial data were also discussed and cited. State-of-the-art review papers were also cited to refer the reader to additional in-depth resources on specific topics.

This paper reviews current data on the classification and functions of monocytes/macrophages; discusses current hypotheses regarding the involvement of monocytes/macrophages in atherogenesis; and highlights existing gaps in evidence. This review is primarily aimed at readers with a background in clinical medicine who are interested in the involvement of monocytes/macrophages in atherogenesis.

## 2. General Characteristics of Monocytes/Macrophages

### 2.1. Monocytes

#### 2.1.1. Monocyte Subpopulations

Monocytes are a type of leukocyte (white blood cell) accounting for 5–10% of the total number of white blood cells present in peripheral blood [7]. Monocytes are derived from hematopoietic cells located in red bone marrow. These monocytes enter the bloodstream, where a portion of them circulate freely in the blood, while others adhere to the marginating pool [8]. This pool accounts for approximately 60% of the monocytes present in the blood. The movement of monocytes between the marginating and circulating pools is a dynamic process influenced by various factors. A significant number of monocytes can be transferred from the marginating pool to the circulating pool within a few minutes, and vice versa [9]. The migration of monocytes to specific bodily tissues is influenced by a variety of chemotactic factors. One such factor is monocyte chemoattractant protein-1 (MCP-1), also known as chemokine C-C motif ligand 2 (CCL2). Following migration, monocytes differentiate into two distinct types of cells—macrophages and dendritic cells [10]—and monocytes that do not migrate into tissues undergo apoptosis [11].

Monocytes constitute a heterogeneous group of cells. During the maturation of leukocytes, proteins specific to different types of leukocytes are synthesized. Flow cytometry allows for the identification of specific molecules, known as clusters of differentiation (CDs), on the surfaces of the membranes of these different types of leukocytes. These markers can be utilized to differentiate between distinct cells. The cell type and degree of cell maturity determine which CDs are located on their surfaces. In essence, CDs function as receptors. Human monocytes are classified based on the expression levels of CD14 and CD16 on their surface [12,13]. CD14 is part of the lipopolysaccharide (LPS) receptor complex, which is a structural component of the external layer of the outer membrane of Gram-negative bacteria [14], while CD16 is a low-affinity receptor for the crystallizable fragment (Fc fragment) of immunoglobulin G (IgG) [15]. There are three primary subpopulations of monocytes: classical (CD14++CD16−), intermediate (CD14++CD16+), and non-classical (CD14+CD16++) [16]. This classification is illustrated in Figure 2. Intermediate and non-classical monocytes are formed sequentially from classical monocytes [17,18,19].

Classical monocytes account for 84–89% of all monocytes circulating in the blood [17,18,19,20,21,22]. In a stable organism (i.e., an organism wherein the stability of the internal environment is maintained), classical monocytes mobilized from the bone marrow remain in the bloodstream for approximately 24 h. Thereafter, 99% of them migrate to tissues or undergo cell death. Approximately 1% of classical monocytes undergo a differentiation process through which they are converted into intermediate monocytes [17]. These monocytes, which are also known as pro-inflammatory monocytes [23], account for 4–7% of all monocytes circulating in the blood [17,18,19,20,21,22] and persist in the bloodstream for approximately four days, after which they undergo differentiation into non-classical monocytes [17]. Non-classical monocytes are smaller than classical and intermediate monocytes [20,21] and account for 7–9% of all monocytes circulating in the blood [17,18,19,20,21,22]. This type of monocyte is characterized by increased maturity and a prolonged circulation time in the blood in contrast to monocytes of other subpopulations, with a mean lifespan of approximately seven days [17].

A distinguishing behavior of non-classical monocytes is their capacity to survey the inner lining of blood vessels [19,24]. These cells are also capable of recognizing and removing damaged and dying cells, their decay products, and various disease-causing factors through a process known as phagocytosis. In contrast to classical monocytes, which roll along the endothelium prior to penetrating the subendothelial matrix, non-classical monocytes possess the capacity to continuously crawl along the surface of the endothelium [21].

#### 2.1.2. Migration of Monocytes into Tissues

Earlier studies hypothesized that monocytes primarily serve to respond to environmental changes, migrate to tissues in response to specific signals, and differentiate into macrophages. Following the discovery of dendritic cells, the potential for monocytes to undergo differentiation was substantiated. As more information was gathered, it was discovered that monocytes have a variety of functions, including the ability to patrol the inner lining of blood vessels and phagocytose, present antigens to T lymphocytes, and participate in angiogenesis [25]. Monocytes secrete various cytokines and chemokines, thereby regulating inflammation and the immune response. Monocytes from distinct subpopulations exhibit different functions depending on whether the body is in a stable state or undergoing a pathological process.

In a healthy body, classical monocytes constantly migrate into tissues to maintain the required levels of macrophages [26]. This information contributes to our understanding of the potential mechanisms underlying the brief existence of classical monocytes within the bloodstream. In the event of systemic inflammation, classical monocytes rapidly migrate to the tissues, where they differentiate into macrophages and dendritic cells [26]. In the event of local inflammation, classical monocytes are the first of the three monocyte subpopulations to migrate into the affected tissues [26]. Following this initial migration, intermediate monocytes enter the tissues. However, it has been hypothesized that their presence in tissues may not be due to migration but rather differentiation from the classical monocytes already present within the tissues. An in vitro study demonstrated that monocytes from all three subpopulations have the ability to differentiate into both classical (M1) and alternative (M2) macrophages [21], with macrophages derived from classical monocytes exhibiting the most pronounced phagocytic activity. The aforementioned study also demonstrated that classical monocytes constitute the primary source of dendritic cells of monocytic origin.

Monocytes are regarded as antigen-presenting cells [27]. This function is primarily linked to intermediate monocytes and predominantly associated with high expression of major histocompatibility complex class II (MHC-II) molecules—which are involved in antigens’ presentation to T lymphocytes—relative to monocytes of other subpopulations [28]. A study conducted using an animal model demonstrated that monocytes that migrate into tissues may not undergo differentiation into macrophages or dendritic cells. These monocytes have been shown to be capable of delivering antigens to lymph nodes for presentation to naive T cells [29]. However, studies have yet to demonstrate such behavior exhibited by monocytes in the human body. Assumptions about the antigen-presenting role of monocytes based primarily on the results of in vitro studies require further confirmation [25].

#### 2.1.3. Assessment of Monocytes’ Functions

One common method for assessing monocyte function is to study their response to various stimuli in a cell culture, particularly the secretion of cytokines in response to LPS stimulation. Studies comparing different monocyte subpopulations are heterogeneous and contradictory in nature. In a frequently cited study conducted by Cros J. et al., it was found in vitro that classical monocytes from healthy individuals produced large quantities of proinflammatory cytokines, including interleukin 8 (IL-8), interleukin 6 (IL-6), chemokine MCP-1, and chemokine C-C motif ligand 3 (CCL3) [20]. Concurrently, classical monocytes produced moderate amounts of the anti-inflammatory cytokine interleukin 10 (IL-10). Intermediate monocytes play a major role in the production of pro-inflammatory cytokines in response to LPS stimulation, including tumor necrosis factor-alpha (TNF-α), interleukin-1 beta (IL-1β), and IL-6. It has been demonstrated that analogous stimulation also results in a substantial increase in the production of the chemokine CCL3 by intermediate monocytes, which also produce moderate quantities of the anti-inflammatory cytokine IL-10. Non-classical monocytes produced minimal pro-inflammatory cytokines in response to LPS stimulation; however, they produced TNF-α, IL-1β, and CCL3 in response to stimulation with viral nucleic acids and immune complexes. This finding suggests the potential involvement of non-classical monocytes in the immune response to viral infection and the pathogenesis of autoimmune diseases. In contrast to the findings reported by Cros J. et al. [20], another study has indicated that non-classical monocytes exhibit a heightened response to LPS stimulation, producing elevated levels of TNF-α and IL-1β compared to other monocyte subpopulations [28]. Concurrently, IL-6 and IL-8 were produced in equivalent levels by monocytes from all three populations. Ong S.M. et al. also found that the production of TNF-α by non-classical monocytes was most pronounced in response to LPS stimulation [30]. The study by Karsulovic C. et al. [31] demonstrated that classical monocytes exhibited the highest production of IL-1β and IL-6 in response to LPS stimulation, simultaneously revealing uniform basal secretion of these cytokines by monocytes of all three subpopulations.

Monocytes’ surfaces are equipped with Toll-like receptors (TLRs), which are pattern recognition receptor (PRRs) [32] that recognize pathogen-associated molecular patterns (PAMPs). These PAMPs include molecules that are an integral component of the bacterial cell wall or bacterial flagellin protein. The binding of TLRs to ligands from microbes that have entered the body leads to the production of cytokines and other molecules associated with inflammation. Monocyte TLRs have also been demonstrated to be capable of recognizing molecules released from damaged or dead cells, known as damage-associated molecular patterns (DAMPs). In response to TLR agonists, all three monocyte subpopulations secrete the pro-inflammatory cytokines IL-1β, IL-6, and TNF-α [21]. Classical monocytes have been shown to produce the highest levels of these cytokines, with the exception of TNF-α, which is secreted in levels comparable to those in classical and intermediate monocytes. Non-classical monocytes have been observed to produce pro-inflammatory cytokines in response to TLR agonists to a lesser extent than monocytes of other subpopulations.

It should be acknowledged that in vitro studies only mirror the actions of monocytes within a living organism to a limited degree. In their natural environments, monocytes exist within a diverse cellular milieu, significantly divergent from the conditions observed in in vitro experiments. Nonetheless, in vitro investigations indicate that there are differences in the functions of leukocytes from different subpopulations. The frequent discrepancies in the results of in vitro studies evaluating the functions of leukocytes from different subpopulations are likely attributable to the varying experimental conditions.

### 2.2. Macrophages

#### 2.2.1. Resident and Monocyte-Derived Macrophages

Macrophages represent an exceptionally diverse group of cells. Depending on their origins, they are classified as either resident macrophages or macrophages formed from monocytes [33,34]. Resident macrophages are formed and distributed throughout all tissues during the prenatal period. In this environment, they remain in a static position and are capable of self-reproduction throughout their entire lifespan [26,35]. Depending on their location, resident macrophages exhibit certain tissue-specific characteristics and are given different names (e.g., Kupffer cells in the liver, alveolar macrophages in the lungs, Langerhans cells in the skin, and microglia in the central nervous system) [36,37]. Resident macrophages are long-lived cells, with the duration of their lifespan ranging from months to years depending on their location [26]. Macrophages of monocytic origin are formed from monocytes that were circulating in the blood after their migration into tissues. The differentiation of monocytes is influenced by the specific tissues to which they migrate. Monocytes have the capacity to differentiate into macrophages that are indistinguishable from resident macrophages [38,39] as well as into any type of tissue-specific macrophage, with the exception of microglia [40]. In a healthy organism, the consumption of resident macrophages in tissues during vital activity is relatively low, and significant migration of monocytes into tissues is not required to maintain the necessary number of macrophages. Moreover, monocytes that have migrated into tissues differentiate into resident macrophages gradually [41]. Therefore, lost macrophages are replaced via the proliferation of neighboring resident macrophages, not monocytes, in a healthy organism. Inflammation is accompanied by a rapid decline in the number of resident macrophages due to their death, a phenomenon that is contingent upon the severity of inflammation. Under such conditions, the depleting pool of resident macrophages is rapidly replaced by macrophages of monocytic origin [42]. The subsequent proliferation of both resident macrophages and macrophages of monocytic origin, as well as their differentiation into resident macrophages, restores the level of macrophages in the tissues [43]. It is hypothesized that when confronted with a pathogen, resident macrophages exhibit a propensity to attract monocytes rather than engage in direct combat. Upon differentiating into macrophages, these monocytes initiate an active response against the pathogen [44]. It has been posited that macrophages of monocytic origin exhibit elevated phagocytic activity in comparison to their resident counterparts [45].

#### 2.2.2. The Phagocytic and Antigen-Presenting Functions of Macrophages

Macrophages, in conjunction with granulocytes (e.g., neutrophils, eosinophils, and basophils) and dendritic cells, are considered “professional” phagocytes. The capacity for phagocytosis is intrinsic to both resident macrophages and macrophages of monocytic origin [34]. Phagocytosis is initiated by the recognition of pathogenic microorganisms via macrophage PRRs [46]. In addition to phagocytosis of external pathogens, macrophages also phagocytose and digest “unnecessary” cells, that is, dead and damaged cells, their decay products, and various harmful agents. The interaction of PRRs with their respective ligands activates a variety of signaling pathways owing to the diversity of these receptors.

Another significant function of macrophages is the presentation of antigens to T lymphocytes (lymphocytes of thymic origin), also known as T cells [46]. T cells are distinguished from other types of lymphocytes by unique T-cell receptors (TCRs) that are located on their surfaces. These receptors are capable of recognizing and interacting with antigenic peptides that are bound to major histocompatibility complex (MHC) molecules present on the surfaces of antigen-presenting cells [47,48]. In the absence of MHC–antigen binding, T cells are unable to recognize an antigen. The two predominant types of T cells are CD4+ T cells and CD8+ T cells. The maturation of T cells from precursor cells is a multi-step process that occurs in the thymus, after which mature naive (antigen-naive) T cells leave the thymus and enter the bloodstream. Subsequently, following interaction with antigens linked to MHC class II (MHC-II) and class I (MHC-I) molecules within the lymph nodes, naive CD4+ T cells and CD8+ T cells recognize antigens and become activated.

In addition to interaction with an antigen, T cell activation necessitates the presence of additional costimulatory signals and interaction with cytokines. The differentiation of CD4+ T cells into distinct T helper (Th-cell) subpopulations, such as Th1, Th2, Th17, and others, is influenced by the interaction of cytokines and costimulatory molecules expressed on antigen-presenting cells. This process also leads to the development of regulatory T cells (Treg cells) [47,49]. The predominant cytokines that regulate the differentiation of Th1 cells are IL-12 and interferon gamma (IFN-γ) [50]. IL-12, secreted by activated antigen-presenting cells, has been shown to play a pivotal role in the differentiation of naive T cells into effector Th1 cells by activating the transcription factor STAT4 [51], which, in turn, has been shown to induce the production of IFN-γ in Th1 cells. IFN-γ has been demonstrated to trigger the differentiation of naive T cells by activating the transcription factors STAT1 and T-bet, which, in turn, induce even higher production of IFN-γ. Consequently, a positive feedback loop is established, thereby stabilizing IFN-γ production. In addition to IFN-γ, Th1 cells have been observed to produce other cytokines, including IL-2, TNF-α, TNF-β, and granulocyte–macrophage colony-stimulating factor (GM-CSF) [52]. The cytokines that most commonly regulate Th2 cell differentiation are IL-4 and IL-2 [51], and activated Th2 cells have been shown to produce IL-4, IL-5, IL-9, IL-10, and IL-13 [52]. The distinction in cytokine production in Th1 and Th2 cells, chiefly in their pivotal cytokines IFN-γ and IL-4, respectively, delineates the predominant direction of the immune response.

After an antigen is absorbed by a macrophage, it is broken down inside the cell. The resulting antigenic peptides bind to MHC-II molecules, and the protein-bound antigens are transferred to the membrane for presentation to Th cells, thus initiating an immune response [47]. It is hypothesized that antigen presentation can occur in non-lymphoid organs [53] and that antigens are presented not to naive T cells but rather to T cells that have already undergone partial activation through prior contact with an antigen within the lymph nodes. The interaction between effector Th1 cells and macrophages results in a significant augmentation in the activation of the latter, and vice versa. Specifically, IFN-γ secreted by Th1 cells stimulates the production of IL-12 by macrophages, which in turn leads to increased secretion of IFN-γ by Th1 cells [53]. It has been posited that IL-4 and IL-13, which are secreted by Th2 cells, may play a role in the development of an anti-inflammatory (M2) macrophage phenotype in vivo [54].

#### 2.2.3. Macrophage Polarization

Inflammation is an essential protective response of the body aimed at eliminating noxious agents, neutralizing their deleterious effects, and initiating the healing process. Macrophages are pivotal in the initiation, maintenance, and resolution of inflammation. An inflammatory response is initiated when cells with PRRs, including resident macrophages, interact with PAMPs or DAMPs. In a healthy body, resident inactive (non-polarized) macrophages engulf apoptotic cells, producing negligible quantities of inflammatory mediators [55]. Upon detection of a pathogen, inactive macrophages (M0 macrophages) undergo classical activation (polarization), resulting in the formation of the classical (inflammatory) M1 macrophage phenotype [56]. This process initiates the production of pro-inflammatory cytokines (Figure 3). The onset of inflammation and tissue damage instigates the migration of classical monocytes from the blood into these tissues, along with their differentiation into M1 macrophages [57]. In addition to interaction with a pathogen, classical activation of monocytes/macrophages is induced by the IFN-γ secreted by Th1 cells. In in vitro studies, M1 polarization of monocyte/macrophage cultures is typically achieved using IFN-γ with a TLR agonist such as LPS [58,59,60]. Using IFN-γ reproduces the phenomenon that occurs when macrophages interact with activated Th1 cells in a living organism to a certain extent. GM-CSF has been shown to promote the differentiation of monocytes and inactive macrophages into the M1 macrophage phenotype [21,61]. GM-CSF, a cytokine, is secreted in large quantities by activated immune system cells when inflammation occurs [62], and increased GM-CSF secretion has been observed to occur under the influence of such pro-inflammatory cytokines as IL-1α, IL-1β, TNF-α, and IL-12.

M1 macrophages have been shown to secrete pro-inflammatory cytokines, including IL-1β, IL-6, and TNF-α [63]. These cytokines are regarded as the primary pro-inflammatory cytokines [64]. IL-1β synthesis is primarily executed by macrophages, endothelial cells, and smooth muscle cells (SMCs). Studies involving animal models have demonstrated that IL-1β inhibition impedes the progression of atherosclerosis [65]. In the CANTOS study, which was conducted on patients with a history of myocardial infarction, the administration of monoclonal antibodies to IL-1β (Canakinumab) resulted in fewer adverse cardiovascular events relative to patients receiving a placebo [66]. TNF-α synthesis is also predominantly executed by macrophages [65]. TNF-α and IL-1β have been demonstrated to induce the transcription of E-selectin and VCAM-1 by endothelial cells [67]. Despite the plethora of studies that have delineated the pivotal role of TNF-α in atherogenesis, there is presently no compelling evidence that the inhibition of TNF-α can decelerate its progression. The synthesis of IL-6 is primarily carried out by macrophages, endothelial cells, and fibroblasts [65]. This substance has been demonstrated to promote the mobilization of leukocytes to the site of inflammation by activating the endothelium, resulting in increased secretion of adhesion molecules and chemokines. The results of the CANTOS study indicate that the beneficial effect of Canakinumab on the course of atherosclerosis is, to some extent, attributable to a decrease in IL-6 levels when the drug is administered [68]. In addition to the secretion of pro-inflammatory cytokines, M1 macrophages secrete CXC chemokines (C-X-C motif chemokine ligands (CXCLs), e.g., CXCL-5, CXCL-9, and CXCL-10) [63], which play an important role in regulating inflammatory and immune responses [69]. A study employing cell culture techniques demonstrated that oxysterols, which form during LDL oxidation, increase the production of IL-8, a chemokine, in macrophages [70]. Furthermore, M1 macrophages have been shown to produce reactive oxygen species (ROS) and nitric oxide (NO), which are involved in the destruction of pathogenic microorganisms [71,72].

In addition to classical activation, monocytes/inactive macrophages have the capacity to undergo alternative activation, which produces an alternative M2 (anti-inflammatory) macrophage phenotype. These macrophages are involved in the resolution of inflammation and tissue repair, performing phagocytosis of cellular debris, regulating the formation of new blood vessels, exhibiting immunosuppressive activity, and playing a key role in the fight against helminths [34,55]. M2 polarization of macrophages in vitro is typically achieved in the presence of IL-4 and/or IL-13 [59,60,73]. This reproduces, to a certain extent, the phenomenon of macrophages interacting with activated Th2 cells in a living organism. Macrophage colony-stimulating factor (M-CSF) has been demonstrated to promote the differentiation of monocytes/inactive macrophages into the M2 macrophage phenotype [21,61,74].

Four main subtypes of anti-inflammatory macrophages were recently identified—M2a, M2b, M2c, and M2d—which differ in their function and expression of specific molecules on their surface [36,75]. Common to all these macrophage subtypes is high production of the anti-inflammatory cytokine IL-10 and transforming growth factor beta (TGF-β) as well as low production of IL-12 [76]. IL-10 is a key anti-inflammatory cytokine that directly inhibits the proliferation of CD4+ T cells and their production of pro-inflammatory cytokines [77]. It also reduces the expression of MHC-II and costimulatory molecules in monocytes/macrophages, thereby indirectly affecting T cell activation, and has been demonstrated to enhance the expression of IL-4 receptors on the surfaces of macrophages [78], thereby increasing their sensitivity to IL-4 and, consequently, M2 activation. TGF-β is involved in the regulation of important cellular functions such as proliferation, differentiation, motility, and apoptosis [79]. Maintaining tissue homeostasis requires a delicate balance between cell proliferation and cell death, both of which are critical processes that are subject to the influence of TGF-β. This cytokine has been demonstrated to possess the capacity to impede proliferation and to elicit apoptosis in a wide range of cell types, and it has also been observed to exert a suppressive effect on the activity of numerous immunocompetent cells and can inhibit the expression of MHC, costimulatory molecules, and IL-12 in macrophages, thereby indirectly affecting T cell activation. Finally, TGF-β has also been shown to inhibit the secretion of TNF-α and NO in macrophages. M2 macrophages have been shown to secrete C-C motif chemokine ligands (CCLs), including CCL17, CCL22, and CCL24 [63].

Macrophages exhibit significant phenotypic plasticity. Depending on the factors affecting macrophages, they can change between M1 and M2 phenotypes [37,59]. Such plasticity is imperative for maintaining the necessary equilibrium between both macrophage phenotypes in response to changes in their environments. An imbalance in the system invariably gives rise to adverse consequences. For example, excessive numbers of M1 macrophages can result in uncontrolled inflammation [80], while excessive numbers of M2 macrophages can contribute to cancer growth, excessive fibrosis, allergic reactions, and an increased risk of infection [81].

M1 and M2 macrophages are two contrasting macrophage phenotypes that have been used in in vitro studies. In these studies, macrophage cultures are exposed to M1- or M2-polarizing agents; subsequently, the production of cytokines and other biologically active molecules in macrophages is evaluated [58,59,73], as are changes in cell markers on the surfaces of these cells [58] and gene expression [82] in response to exposure to polarizing agents. Alterations in cytokine production, among other phenomena, are not only contingent on the agents utilized and their respective quantities but also on the duration of exposure [59]. It is important to acknowledge that the administration of the same agent to humans and animals does not always result in the same outcomes [60]. The distinction between two opposite macrophage phenotypes is largely arbitrary. In a living organism, macrophages are exposed to a variety of factors; therefore, it is assumed that there are additional macrophage phenotypes, but these have not been sufficiently studied [36].

### 2.3. Dendritic Cells

Dendritic cells have the capacity to present antigenic peptides to T lymphocytes and are the most effective antigen-presenting cells [46]. They have been shown to bind to antigens, firmly fixing them on their surfaces; perform endocytosis; and partially cleave antigens enzymatically. Subsequently, the resulting antigenic peptides bind to MHC molecules, and the bound antigens are transferred to the cell membrane. Dendritic cells can also enter the circulation and migrate to the lymph nodes, where they present antigens to T lymphocytes for recognition. Dendritic cells exhibit heterogeneity in their composition; there are three primary subpopulations: plasmacytoid dendritic cells, myeloid/conventional type 1 dendritic cells, and myeloid/conventional type 2 dendritic cells [83,84]. In addition to these subpopulations, inflammatory dendritic cells have been identified, which are derived from monocytes during periods of inflammation [85]. In vitro studies have shown that GM-CSF, when used in conjunction with other cytokines such as IL-4, can induce the differentiation of monocytes into dendritic cells [86]. Monocyte-derived dendritic cells can differentiate into two distinct subtypes: (1) dendritic cells that promote inflammation and immune cell activation by presenting antigens to T cells and secreting pro-inflammatory cytokines [87] and (2) dendritic cells that promote immune tolerance by presenting antigens to T reg cells and secreting anti-inflammatory cytokines [88]. Dendritic cells’ role in atherogenesis is explored in detail elsewhere [89,90].

## 3. Participation of Monocytes/Macrophages in Atherogenesis

### 3.1. Macrophages’ Uptake of Modified Low-Density Lipoproteins (LDLs) in the Arterial Walls

The progression of atherosclerosis and its associated complications is primarily instigated by the infiltration of cholesterol-rich lipoproteins containing apolipoprotein B (apoB) into the subendothelial matrix, where they are retained [91,92]. The “response-to-retention” model is a theoretical framework that models the development of atherosclerosis, with the primary initiating factor identified as the retention of apoB-containing lipoproteins. For many years, there has been debate about how low-density lipoproteins (LDLs) penetrate the subendothelium. In recent years, the concept of LDLs crossing the endothelium via transcytosis, i.e., the transcellular transport of various macromolecules, has been confirmed [93]. During transcytosis, the transported object is recognized by a receptor and penetrates the cell via endocytosis. It is then transported to the opposite cell wall inside the vesicle and then exits into the extracellular space via exocytosis. The presence of small Ω-shaped invaginations on the plasma membranes of endothelial cells (caveolae) has been observed, and their main structural component, caveolin 1 protein, has been identified. In addition, receptors within these invaginations, namely, activin-like kinase receptor 1 (ALK1) and scavenger receptor B1 (SR-B1), which have been shown to bind to LDLs, have been identified as key regulators of their transcytosis [94,95,96]. Despite significant advancements in our understanding of the mechanisms underlying the transportation of LDL into the intima, the physiological significance of this process remains to be fully elucidated [95]. Moreover, the precise mechanisms by which hypercholesterolemia facilitates the penetration of LDLs into arterial walls remain obscure. In addition, the precise mechanisms responsible for the retention of apoB-containing lipoproteins in the intima are not yet fully elucidated. It has been postulated that lipoproteins are retained in the intima due to the interaction of proteoglycans of the subintimal extracellular matrix with the apoB-100 component of lipoproteins [93,97].

LDLs that remain in the interstitial space are susceptible to various oxidative and non-oxidative modifications [93,97]. Modified LDLs have been shown to bind to a variety of scavenger receptors, including CD36, SR-A1, SR-B1, and LOX-1, as well as to TLRs located on the surfaces of macrophages [97,98,99]. This interaction leads to the uptake of LDLs by macrophages, which subsequently lyse them. Receptor-mediated endocytosis has been identified as the predominant mechanism underlying modified LDL uptake [100]. There is evidence that CD36 and SR-A receptors predominantly interact with modified LDLs in their uptake by macrophages [101].

### 3.2. Inflammatory Activation of Macrophages and Their Interaction with the Endothelium

#### 3.2.1. Interactions Between Monocytes/Macrophages and the Endothelium

The process of macrophage activation in atherosclerosis is characterized by the activation of various signaling pathways. These pathways consist of chains of sequential biochemical reactions that arise when intracellular molecules interact during the transmission of an external signal from the cell surface to the nucleus, resulting in a change in macrophage function [63,102]. These signaling pathways include the NLRP3 inflammasome activation, TLR, HIF-1α, NF-κB, JAK/STAT, TBK/IRF3, and PI3K-Akt/PKB signaling pathways. The internalization of modified LDLs by resident macrophages predominantly results in their inflammatory activation [103]. Specifically, the internalization of oxidized LDLs by macrophages prompts the activation of a multimeric cytosolic protein complex, known as the NLRP3 inflammasome. This results in elevated secretion of pro-inflammatory cytokines, including IL-1β and IL-18, by macrophages [104]. In addition to the uptake of oxidized LDLs, the NLRP3 inflammasome is activated by the uptake of cholesterol crystals by macrophages. Cholesterol crystals may be present in atherosclerotic lesions even in the early stages of their development [105,106]. Inhibition of the NLRP3 inflammasome has been demonstrated to suppress the secretion of IL-1β and IL-18 in macrophages [102], while the interaction of oxidized LDLs with the Toll-like receptor TLR4 instigates intracellular processes, resulting in increased IL-6 secretion by macrophages [107]. Oxidized LDLs have been demonstrated to elicit an intracellular signaling pathway, the central component of which is the pro-inflammatory transcription factor nuclear factor kappa-light-chain-enhancer of activated B cells (NF-κB), playing a pivotal role in the regulation of the genes involved in inflammatory and immune responses [108].

Inhibition of NF-κB leads to a decrease in the production of TNF-α, IL-18, IL-1β, and IL-6. The binding of oxidized LDLs to CD36 receptors has been demonstrated to increase the secretion of vimentin, a protein of intermediate filaments (thread-like structures that are part of the cytoskeleton), by macrophages into the extracellular space [109]. Once in the extracellular space, vimentin has been shown to induce TNF-α secretion by macrophages and enhance the production of TNF-α and IL-6 mediated by oxidized LDLs. Oxidized LDLs have been shown to increase the expression of the cell enzyme gene receptor-interacting protein kinase 1 (RIPK1) in macrophages, which triggers a signaling pathway involving NF-κB, leading to increased secretion of IL-1α and IL-1β [110].

The secretion of pro-inflammatory cytokines by classically activated macrophages leads to the activation of endothelial cells [111,112]. The binding of TNF-α and IL-1β, which are secreted by macrophages, to endothelial cell receptors triggers a signaling pathway involving the pro-inflammatory transcription factor NF-κB [112]. Furthermore, endothelial cell activation is induced by the binding of oxidized LDLs to the lectin-type oxidized LDL receptor 1 (LOX-1) on the surfaces of endothelial cells [113]. Activation of these cells has been demonstrated to induce the secretion of various chemokines, including IL-8 (CXCL8), CCL5 (regulated upon activation, normal T cell expressed and secreted, also known as RANTES) [114], and MCP-1 (CCL2). This process also results in the expression of cell adhesion molecules on the surfaces of endothelial cells, such as vascular cell adhesion molecule 1 (VCAM-1), intercellular adhesion molecule-1 (ICAM-1), E-selectin, and P-selectin. Furthermore, activated endothelial cells secrete inflammatory mediators, including the pro-inflammatory cytokines IL-1 and IL-6 [34,113,115]. A notable chemokine that plays a crucial role in attracting monocytes to the site of inflammation is MCP-1, which is secreted by the activated endothelium [116]. Its interaction with receptors on the surfaces of monocytes has been shown to trigger the recruitment of monocytes to the activated endothelium (Figure 4) [117,118]. Furthermore, selectins presented by the activated endothelium interact with monocytes, reducing their velocity and allowing them to overcome the forces of blood flow and begin rolling along the endothelium [119,120]. P-selectin is stored in the Weibel–Palade bodies of endothelial cells, where, in the presence of pro-inflammatory stimuli, it can rapidly move to the surface. The expression of E-selectin by endothelial cells is strictly regulated by pro-inflammatory cytokines such as TNF-α and IL-1β [121]. In contrast, these cytokines do not regulate P-selectin activity in humans. E-selectin facilitates monocyte rolling at a significantly lower rate than P-selectin. In light of these observations, E-selectin’s role in attracting effector cells to the developing focus of inflammation is considered to be of paramount importance [122].

#### 3.2.2. Monocytes’ Migration into the Subendothelial Matrix

It has been demonstrated that VCAM-1 and ICAM-1, ligands for transmembrane monocyte integrin receptors, are critical factors in the adhesion of monocytes to the endothelium [120]. The binding of VCAM-1 and ICAM-1 to integrins necessitates their activation, which is facilitated by chemokines immobilized on the endothelial surface [122,123]. The subsequent migration of monocytes into the subendothelial matrix occurs both transcellularly and paracellularly [34]. The binding of ICAM-1 to the monocyte integrins also instigates the phosphorylation of VE-cadherin, a cell adhesion protein situated at the junctions of endothelial cells. This process enables monocytes to migrate between endothelial cells. It is hypothesized that VCAM-1 plays a pivotal role in the initiation of atherosclerosis [124].

The majority of monocytes that migrate into the subendothelial matrix are classical monocytes [125]. After entering the intima, monocytes undergo differentiation, which ultimately results in their transformation into macrophages or dendritic cells. In the developing inflammatory focus, monocytes that differentiate predominantly into M1 macrophages exhibit a high phagocytic capacity. Their uptake of modified LDLs instigates the secretion of pro-inflammatory cytokines, contributing to the inflammatory response.

In the initial stage of the formation of atherosclerotic lesions, macrophages accumulate in the vascular wall due to the recruitment of monocytes from the bloodstream. Over time, this accumulation causes macrophage proliferation [126]. This phenomenon is contingent, albeit not exclusively, on the local production of colony-stimulating factor 1 (CSF1) by the endothelium and SMCs [127]. A deficiency in CSF1 production by these cells will result in a significant decrease in macrophage proliferation.

#### 3.2.3. The Endothelial–Mesenchymal Transition in the Focus of Atherosclerotic Lesions

The occurrence of chronic inflammation in atherosclerotic lesions leads to the gradual transformation of endothelial cells into cells with a phenotype characteristic of mesenchymal cells and, consequently, to the development of endothelial dysfunction [128,129]. This process has been termed the endothelial–mesenchymal transition, in which TGF-β is regarded as a pivotal element [130]. TGF-β is a cytokine that performs a variety of functions, including stimulation of cell differentiation and proliferation. It has been established that normal, non-activated endothelial cells exhibit minimal expression of TGF-β type 1 receptors, a characteristic that renders these cells almost entirely impervious to the effects of TGF-β. Inflammation of the vascular wall leads to a rapid increase in the expression of these receptors and, consequently, induces the endothelial–mesenchymal transition. The upregulation of TGF-β type 1 receptor expression during inflammation is tightly associated with the downregulation of the expression of fibroblast growth factor receptor 1 (FGFR1) in endothelial cells. In atherosclerosis, a decrease in the expression of FGFR1 receptors has been observed; it has been attributed to the action of pro-inflammatory mediators, including IFN-γ, TNF-α, and IL-1β. The transformation of endothelial cells augments vascular wall permeability, modifies the extracellular matrix, and increases the expression of cell adhesion molecules on their surfaces. These processes underpin and amplify the inflammatory response. The number of endothelial cells that have undergone the endothelial–mesenchymal transition in the focus of atherosclerotic lesions is directly correlated with the instability of the atherosclerotic plaque [131].

### 3.3. Formation of Foam Cells

In macrophages, all lipoprotein components are subject to hydrolysis by lysosomal enzymes. The hydrolysis of esterified cholesterol in LDLs results in the formation of free cholesterol, which undergoes partial re-esterification in the endoplasmic reticulum and is subsequently deposited in the cytoplasm as lipid droplets [132]. The remaining free cholesterol is subsequently removed from cells by ABC transporters (ATP-binding cassettes)—membrane proteins that facilitate the transportation of various substrates across cell membranes by utilizing the energy derived from the hydrolysis of adenosine triphosphate (ATP) [133]. The excessive lipid accumulation in macrophages results in their transformation into foam cells, whose abundance increases progressively over time. Foam cells are a hallmark of atherosclerosis, and they are part of fatty streaks, the initial visible manifestation of the disease. The primary source of foam cell formation is macrophages, which are derived from monocytes circulating in the blood [76] (Figure 5). In addition, a significant number of foam cells are formed from SMCs of the arterial wall. These foam cells, like macrophages, capture modified LDLs [76,134]. SMCs have been demonstrated to transdifferentiate into macrophage-like cells during atherogenesis [135]. The aspects of macrophage and SMC interaction in foam cell formation are comprehensively outlined in the review by Xiang et al. [136]. Other sources of foam cells may include stem progenitor cells and endothelial cells, although little is known about the extent of their contribution to the resulting foam cell pool [137,138]. The results of (mainly in vitro) studies on the production of pro-inflammatory cytokines by foam cells are inconsistent. These studies are reviewed elsewhere [139]. A number of studies have confirmed the production of such cytokines, while others have revealed a decline in this production. Foam cells have been demonstrated to secrete matrix metalloproteinases (MMPs), which are enzymes that degrade the proteins that comprise the extracellular matrix. This process can result in the destabilization and subsequent rupture of the atherosclerotic plaque [140].

### 3.4. Formation of Atherosclerotic Plaques

#### 3.4.1. Macrophages in the Atherosclerotic Plaque Core

As atherosclerosis progresses, atherosclerotic plaques form in areas where fatty streaks are located. Atherosclerotic plaques (atheromas and fibroatheromas) are local formations within the artery wall, comprising a lipid core that is enveloped by a fibrous capsule. The core of the plaque contains a variety of components, including macrophages, foam cells, T lymphocytes, cellular debris, lipids, cholesterol crystals, and calcium. The fibrous cap consists of SMCs and the extracellular matrix that is infiltrated with lymphocytes and macrophages [141]. As atherosclerosis progresses, a significant number of foam cells and macrophages undergo necrosis, contributing to the formation and subsequent growth of the lipid core. Excessive accumulation of lipids in macrophages disrupts their intracellular metabolism, triggering inflammatory signaling pathways, endoplasmic reticulum stress, and, ultimately, macrophage death. The mechanisms and pathways of macrophage death in atherosclerosis have been thoroughly described in other reviews [141,142,143]. A brief discussion of cell death mechanisms essential for atherosclerotic plaque development and destabilization is provided below.

#### 3.4.2. Macrophage Death Mechanisms in Atherosclerosis: Apoptosis

Macrophages have been observed to undergo a variety of cell death mechanisms, including apoptosis, necroptosis, pyroptosis, and ferroptosis. Apoptosis is a type of programmed cell death that eliminates “unnecessary” cells from the body. Two primary pathways in the process of apoptosis have been identified: the external receptor-dependent pathway and the internal mitochondrial-dependent pathway. The external pathway involves the binding of ligands, such as TNF-α, TRAIL (TNF-related apoptosis-inducing ligand), and FasL (ligand for the membrane molecule Fas), to their receptors located on the cell membrane. This binding triggers activation of resting proenzymes (pro-caspases) in the cytoplasm, which are subsequently converted into active enzymes, known as effector caspases, which cleave cellular proteins [144]. The mitochondrial pathway is initiated by intracellular homeostasis disruption, leading to an increase in the permeability of the outer mitochondrial membrane. This leads to the release of intermembrane mitochondrial proteins, particularly cytochrome C, into the cytosol. This event initiates intracellular processes that culminate in the formation of effector caspases. The Bcl-2 family of proteins plays a central role in mitochondrial-dependent apoptosis. Depending on their structures, these proteins can either promote or inhibit the onset of apoptosis. The increased permeability of the outer mitochondrial membrane allows the proapoptotic proteins Bax and Bak to exert their influence on the membrane. Apoptosis is characterized by the formation of apoptotic bodies surrounded by a membrane, which prevents toxic and immunogenic debris from entering the intercellular space. Apoptotic bodies are recognized, absorbed, and digested by macrophages; those that are not removed undergo secondary necrosis.

In the atherosclerotic context, macrophages undergo apoptosis under the influence of numerous factors, including oxidative stress, hypoxia, elevated cytokine concentrations (e.g., IFNγ), and cholesterol overload [145]. The induction of apoptosis necessitates the concerted action of an inducing factor that is sufficiently pronounced and prolonged and frequently attributable to a multifaceted etiology rather than a solitary agent. Macrophage apoptosis occurs at all stages of atherosclerotic plaque formation [146].

In the early stages of atherogenesis, macrophage apoptosis, in conjunction with phagocytosis of the resultant apoptotic bodies and the migration of macrophages out of the lesion, serves to curtail macrophage accumulation within the lesion, thereby impeding atherosclerotic lesion growth. In later stages, the combination of apoptosis and the impaired clearance of apoptotic bodies contributes to the formation of a necrotic core consisting of dead foam cells, macrophages, SMCs, lymphocytes, and extracellular matrix [147,148]. These contents have been demonstrated to be a source of pro-inflammatory cytokines, proteases, and DAMPs, which enhance inflammation. High-mobility group box 1 (HMGB1) protein is among the most thoroughly studied DAMPs in the context of atherosclerosis, with macrophages identified as a particularly noteworthy source of HMGB1 [149]. Upon its entry into the extracellular matrix, HMGB1 binds to various receptors and upregulates the release of pro-inflammatory cytokines, including TNF-α, IL-1, IL-6, and IL-8, by macrophages [144]. An experiment involving an animal model demonstrated that neutralization of HMGB1 inhibits atherogenesis, particularly via reducing macrophage migration [150].

#### 3.4.3. Macrophage Death Mechanisms in Atherosclerosis: Necroptosis

In addition to apoptosis, macrophages are subjected to other types of programmed cell death. Necroptosis is initiated by the interaction of TNF-α with the TNFR1 receptor on the macrophage cell membrane. This results in the formation of a specific protein complex that activates intracellular processes, which trigger the formation of a necrosome complex consisting of RIPK1 and RIPK3 proteins (Figure 6) [151]. This necrosome complex activates the MLKL protein, which approaches the cell membrane and disrupts its integrity by forming pores. This ultimately results in macrophage death [141]. Necroptosis can also be triggered by the activation of other receptors, such as TLRs (TLR4 and TLR3), or by the inhibition or inactivation of caspases [151]. Necroptosis is characteristic of the later stages of atherogenesis and results in release of the intracellular contents into the extracellular matrix, instigating a pronounced inflammatory response.

#### 3.4.4. Macrophage Death Mechanisms in Atherosclerosis: Pyroptosis

Pyroptosis is induced by the NLRP3 inflammasome, which activates caspase-1 protease, thereby initiating the canonical caspase-1-dependent pathway (Figure 7) [141,145]. The NLRP3 inflammasome serves to produce pro-inflammatory cytokines IL-1β and IL-18 from their precursor molecules and partially cleaves gasdermin D, leading to the release of the N-domain of gasdermin D, which results in the formation of pores in the cell membrane. This results in osmotic swelling and cell death. The release of cytokines IL-1β and IL-18, as well as DAMPs, through the pores into the extracellular space augments inflammation. In addition to the NLRP3 inflammasome, the AIM2 protein can also facilitate pyroptosis. Pyroptosis is characteristic of advanced atherosclerotic lesions.

#### 3.4.5. Macrophage Death Mechanisms in Atherosclerosis: Ferroptosis

Ferroptosis occurs due to peroxidation of polyunsaturated fatty acids in the presence of iron ions. This leads to rupture of the cell membrane and release of the intracellular contents into the extracellular space (Figure 8) [141,145]. The peroxidation of polyunsaturated fatty acids can occur via an enzymatic (canonical) pathway or a non-enzymatic (non-canonical) pathway, which occurs via the Fenton reaction. Disrupted activity of glutathione peroxidase-4 (GPX4) and the accumulation of a labile pool of divalent iron ions in the cytoplasm predisposes cells to ferroptosis. Intraplaque hemorrhage and phagocytosis of red blood cells by macrophages largely contribute to increases in this labile iron pool [152].

#### 3.4.6. Fibrous Cup Formation

As an atherosclerotic lesion advances, a fibrous cup forms around the initially lipid and subsequently necrotic core of the plaque. This cup consists of SMCs surrounded by the extracellular matrix with high collagen and elastin fiber content [153]. In the human body, the fibrous cup is initially composed of the same material as the original intima. As the disease advances, the cup’s structure becomes saturated with SMC collagen fibers. These SMCs migrate from the media into the fibrous cap and switch from the contractile phenotype to the synthetic phenotype. They lose their ability to produce contractile proteins that are integral to these cells, instead acquiring a greater ability to produce extracellular matrix. SMCs exhibit a high degree of plasticity, enabling them to swiftly modify their phenotype in response to external stimuli. It is hypothesized that among various factors, the production of certain cytokines by endothelial cells, macrophages, and T lymphocytes may influence the migration, proliferation, and phenotype switching of SMCs [154].

The fibrous cap of a stable atherosclerotic plaque is separated from the lumen of the artery by the endothelium. During a lesion’s development, there is potential for plaque rupture, leading to the formation of an occluding thrombus. The thickness of the fibrous cap is the most significant factor in determining the likelihood of its rupture. The thinning of the fibrous cap predisposes it to rupture and is widely regarded as a sign of plaque vulnerability. Macrophages and foam cells are the primary sources of MMPs, a group of enzymes capable of cleaving virtually all components of the extracellular matrix [155]. Increased MMP activity has been shown to decrease the density of the fibrous cap and thereby increase the likelihood of plaque rupture [156,157].

#### 3.4.7. Fibroblasts in the Atherosclerotic Plaque

The endothelial–mesenchymal transition and contractile-to-synthetic phenotype switching of SMCs are considered the main sources of cells with the synthetic phenotype in the atherosclerotic plaque [158]. Nonetheless, recent findings indicate that fibroblasts may also be involved in atherogenesis from its earliest stages [159]. Fibroblasts are heterogeneous cells of mesenchymal origin, located in adventitia of healthy vessels, where they reside in a quiescent state [160]. In response to numerous pro-inflammatory stimuli, such as TGF-β, IL-10, platelet-derived growth factor (PDGF), vascular endothelial growth factor (VEGF), epidermal growth factor receptor (EGFR), and amphiregulin (AREG) released by macrophages, fibroblasts become activated [161]. It was postulated that neovascularization, observed in atherosclerotic lesions, can facilitate the delivery of fibroblasts into the plaque [160]. Nonetheless, it is not clear to which extent adventitious fibroblasts contribute to the pool of cells with the synthetic phenotype in atherosclerotic lesions. The high heterogenicity and plasticity of fibroblasts, as well as the ability of other types of cells to transform into fibroblast-like cells, hinder the clear distinction of fibroblasts and fibroblast-like cell subsets in the atherosclerotic plaque. In addition, macrophages can potentially differentiate into fibroblast-like cells and produce components of extracellular matrix, including collagen fibers [162]. The macrophage-to-myofibroblast transition is a phenomenon observed in fibrotic diseases [163]. It is not clear whether this transition occurs in atherosclerosis.

In addition to components of extracellular matrix, activated fibroblasts produce proinflammatory cytokines, including IL-1, TGF-β, IL-6, reactive oxygen species, and growth factors such as VEGF, maintaining a proinflammatory microenvironment in the atherosclerotic lesions [164]. Different subsets of activated fibroblasts produce a number of MMPs and tissue inhibitors of MMPs (TIMPs), contributing to atherosclerotic plaque stabilization and destabilization [164].

It was postulated that fibroblasts play an important role in plaque calcification [165] and neointimal growth after coronary artery stenting [166].

### 3.5. Macrophages and the Local Immune Response

The detection of T cells and B cells in atherosclerotic lesions led to the assumption of an autoimmune response being involved in atherosclerosis. Among T cells, CD4+ T cells are the most prevalent in atherosclerotic plaques [47,167]. The capacity of subpopulations of these cells to generate a distinct spectrum of cytokines leads to the manifestation of both atherogenic and antiatherogenic effects. Th1 cells are the most prevalent subpopulation of CD4+ T cells in atherosclerotic lesions. These cells secrete proinflammatory cytokines such as IFN-γ, TNF-α, and IL-2, thereby contributing to the maintenance of chronic inflammation in the arterial wall [168]. IFN-γ is a classic macrophage-activating factor, and its production in Th1 cells is primarily regulated by the transcription factor T-bet. It induces the secretion of pro-inflammatory cytokines and cytotoxic molecules by macrophages as well as the expression of genes that regulate lipid uptake. This, in turn, has been shown to influence the formation of foam cells [167]. IFN-γ has been demonstrated to stimulate the production of chemokines and cell adhesion molecules by endothelial cells, thereby facilitating the migration of monocytes into the intima. It has also been demonstrated that IFN-γ can induce apoptosis in macrophages and stimulate MMP synthesis in these cells, thereby affecting the stability of the atherosclerotic plaque. It was postulated that autoantigens arising from modifications of apoB-100 in LDLs (including oxidation), alterations in the structure of extracellular matrix proteins, and other modifications of various proteins may trigger an immune response in atherosclerosis [167].

In contrast to Th1 cells, Treg cells impede atherogenesis by curtailing the proliferation of proinflammatory Th cells and secreting anti-inflammatory cytokines IL-10 and TGF-β [47,169,170]. During atherogenesis, certain Treg cells can undergo a transformation into a state that is both cytotoxic and pro-inflammatory [171]. The precise mechanisms underlying this transformation of Treg cells remain to be elucidated. The role of other T helpers in atherogenesis, as well as that of CD8+ T cells, is less clear, and the results of studies are contradictory [47,169].

B cells play a pivotal role in both innate and acquired immunity due to their capacity to produce antibodies and secrete cytokines [49,172]. They are distinguished from other types of lymphocytes by their unique surface expression of B cell receptors (BCRs), which enable direct recognition and binding of antigens. The maturation of B cells from precursor cells is a multistage process that is completed in the spleen or regional lymph nodes. Their role in atherogenesis remains a relatively understudied area. In contrast to T cells, B cells are present in low numbers in atherosclerotic lesions. Within the population of B cells found in atherosclerotic plaques, three distinct populations can be identified: B1 cells, B2 cells, and regulatory B (Breg) cells. The various subpopulations within this group exhibit distinct functions in atherogenesis. B1 cells, which include CD5+B1a and CD5−B1b cells, are known to produce immunoglobulin M (IgM). IgM has been observed to bind oxidized LDLs and apoptotic cells, thereby impeding their uptake by macrophages and exerting an antiatherogenic effect [173,174]. Within the B2 cell population, two distinct subpopulations are identified: follicular B2 cells and marginal zone B2 cells (MZ B cells). Follicular B2 cells have been shown to have an atherogenic effect due to the production of IgG and the activation of Th1 cells, causing them to secrete pro-inflammatory cytokines [174,175], while MZ B cells have been shown to exert an antiatherogenic effect through the production of IgM and suppression of Th cells [49,174]. The antiatherogenic effect of Breg cells is attributable to their capacity to secrete the anti-inflammatory cytokine IL-10 and to induce Treg cells [49].

### 3.6. Macrophages and Resolution of Inflammation

Inflammation is a protective response of the body aimed at eliminating the pathogen that caused it. An adequate inflammatory response requires strict control to ensure the necessary intensity and duration of the response. The resolution of inflammation is initiated when the inflammatory response reaches its peak. The ideal outcome of this process is the complete reparation of the structure and function of damaged tissues. Inflammation resolution is an active, well-coordinated process that is controlled by specific endogenous signaling molecules. These molecules include compounds with various chemical structures that exhibit regulatory properties. The resolution of inflammation is based on the inhibition of the migration of polymorphonuclear leukocytes (PMNs) to the site of inflammation; PMN apoptosis; efferocytosis of apoptotic PMNs; inhibition of the migration of monocytes to the site of inflammation and their differentiation into M1 macrophages; a change in the macrophage phenotype from M1 to M2; differentiation of monocytes at the site of inflammation into M2 macrophages; catabolism of pro-inflammatory cytokines; and production of anti-inflammatory mediators [176]. Without an effective and timely resolution, inflammation becomes chronic. The inability to resolve inflammation (or the disruption of the mechanisms) is one of the key characteristics of atherosclerosis [114,177].

Macrophages have been found to play a pivotal role in the resolution of inflammation [178,179], which is marked by the emigration of these macrophages from the site of inflammation. This migration can occur through reverse transendothelial migration or via lymphatic vessels located in the adventitia near the inflammatory focus [114,180]. Macrophage emigration is impaired in atherosclerosis. Reverse transendothelial migration of macrophages has been observed in its early stages, but this process is suppressed as the disease advances [181]. Endothelial adhesion molecules have a negative effect on macrophage emigration [114]. LDL modification also impedes the resolution of inflammation. It has been demonstrated that interaction of oxidized LDLs with the CD36 scavenger receptor of macrophages elicits a signaling pathway that impedes the migratory capacity of macrophages [182]. Efferocytosis, defined as the recognition, engulfment, and destruction of apoptotic cells within the inflammatory focus, is a type of phagocytosis that plays an important role in the resolution of inflammation [183]. Dead cells are engulfed prior to the disruption of their membranes, which impedes the release of intracellular contents into the tissues and mitigates the inflammatory response. Macrophages are instrumental in the process of efferocytosis, which results in the reduced production of pro-inflammatory cytokines and chemokines by macrophages that have engulfed dead cells [184]. There is evidence that efferocytosis is impaired in atherosclerosis [184,185,186,187].

## 4. Monocyte/Macrophage-Directed Therapeutics in Atherosclerosis

Current anti-inflammatory therapeutics for atherosclerosis studied in randomized clinical trials specifically target IL-1β (canakinumab in the CANTOS trial [66]), IL-6 (ziltivekimab in RESCUE, ZEUS [188], and clazakizumab in the POSIBIL6ESKD trials [189]) or have a broader anti-inflammatory function, as with colchicine (as shown in the COLCOT [190], LoDoCo2 [191], COPS [192], CLEAR-SYNERGY [193], and CONVINCE [194] trials) and methotrexate (the CIRT trial [195]). Of these agents, the IL-1β inhibitor canakinumab proved to be effective at MACE rate reduction in patients with previous myocardial infarction and elevated C-reactive protein levels. Furthermore, colchicine treatment produced a significant reduction in the MACE rate in stable coronary artery disease, whereas the results were inconclusive in acute myocardial infarction or ischemic stroke. Moreover, methotrexate treatment did not reduce MACE rates. The trials evaluating anti-IL-6 agents are still ongoing.

Anti-IL-1β and anti-IL-6 agents target downstream links of inflammatory pathways and do not directly target monocytes/macrophages. Nonetheless, there is recent evidence that colchicine can affect monocytes/macrophages. An ex vivo experiment demonstrated that colchicine reduced the proliferative activity of macrophages within human atherosclerotic plaques [196]. It also downregulated CD36 expression, decreasing macrophage lipid uptake and macrophage-to-foam cell transformation [197]. Colchicine also impeded circulating monocyte recruitment and adhesion molecule expression on monocytes in a murine model [198].

To date, monocyte-directed treatment has been studied in vitro and in animal models. New approaches, such as the use of macrophage-targeted nanoparticles or proteolysis-targeting chimera (PROTAC) technology, have been designed to target M1-to-M2 macrophage polarization [199]. For example, a murine model has been used to study the reprogramming of monocytes into adopting an anti-inflammatory state using 4-phenylbutyric acid-mitigated atherogenesis [200]. Pioglitazone, a common anti-diabetic medication, was shown to regulate macrophage polarity towards the M2 phenotype in murine models [201,202]. Another anti-diabetic drug, gliclazide, reduced atherosclerotic plaque burden in a rabbit model. Gliclazide treatment was also associated with anti-inflammatory macrophage polarization, which was ascribed to the anti-NLRP3 inflammasome effect of this drug [203]. A meta-analysis of 14 studies testing 11 different anti-MCP-1 agents demonstrated that these substances attenuated atherogenesis in murine models and reduced macrophage accumulation in atherosclerotic lesions. Nonetheless, the authors indicated a lack of rigorous methodology in the majority of the studies [204]. In another study, an oxidized phospholipid molecule VB-201 hindered atherogenesis in a murine model [205]. In vitro, oligomeric proanthocyanidin reduced monocyte-to-macrophage differentiation induced by oxidized LDL and M-CSF [206]. Pro-efferocytic nanoparticles bearing an inhibitor of the antiphagocytic CD47-SIRPα signaling axis accumulated within lesional macrophages and reactivated phagocytosis, reducing atherosclerotic burden in a murine model [207]. A study by Siegel et al. demonstrated that P2Y12 receptor inhibitors, conventionally prescribed to patients with myocardial infarction and coronary artery stenting, can suppress macrophage migratory functions as well as adhesion of platelet–monocyte aggregates to ICAM-1 [208]. An in-depth review of studies of monocyte-targeting agents can be found elsewhere [76].

## 5. Conclusions

Monocytes/macrophages are critical players in all phases of atherogenesis. The initial phase of atherosclerosis is characterized by the accumulation and modification of LDLs within the subendothelial matrix, followed by the uptake of modified LDL particles by macrophages, leading to the transformation of these macrophages into foam cells. Macrophages produce numerous biologically active substances, including cytokines, lipo- and proteolytic enzymes, and free radicals, that play important roles in atherogenesis. The functional states of macrophages, therefore, play a pivotal role in this process. Atherosclerosis is characterized by the presence of local, chronic, low-grade inflammation in the arterial wall, primarily in areas of the arterial bed that are predisposed to this disease, and macrophages have been identified as playing a key role in the initiation and perpetuation of inflammation within the vascular wall. During atherogenesis, macrophages interact with the extracellular matrix and all cells in the vascular wall, including other cells of the immune system. They also participate in the progression of atherosclerotic plaques and contribute to their destabilization and play a pivotal role in resolving inflammation.

## Figures and Tables

**Figure 1 ijms-26-10962-f001:**
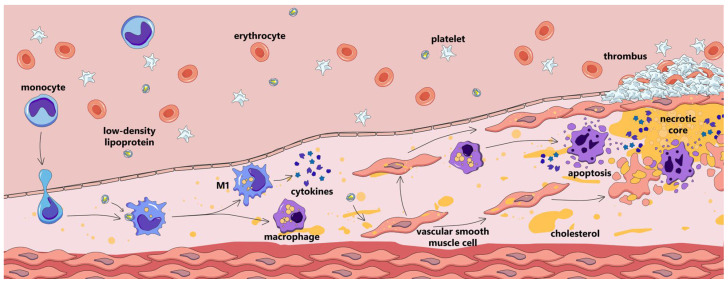
Atherosclerotic plaque development.

**Figure 2 ijms-26-10962-f002:**
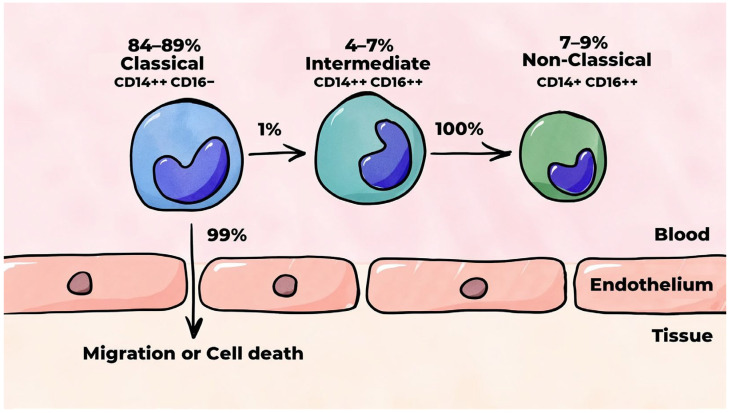
Human monocyte subpopulations.

**Figure 3 ijms-26-10962-f003:**
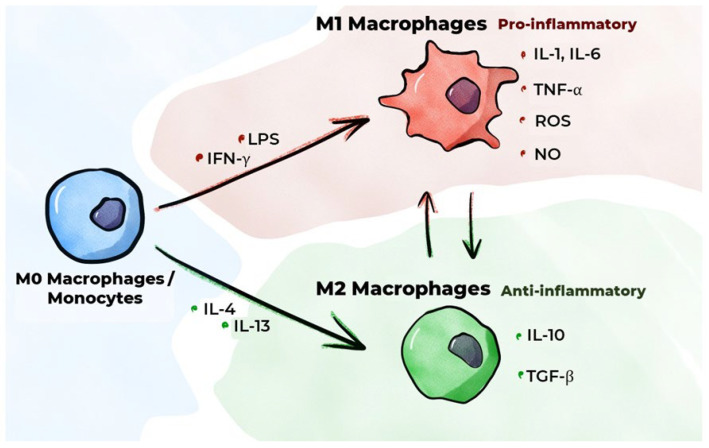
M1/M2 macrophage polarization.

**Figure 4 ijms-26-10962-f004:**
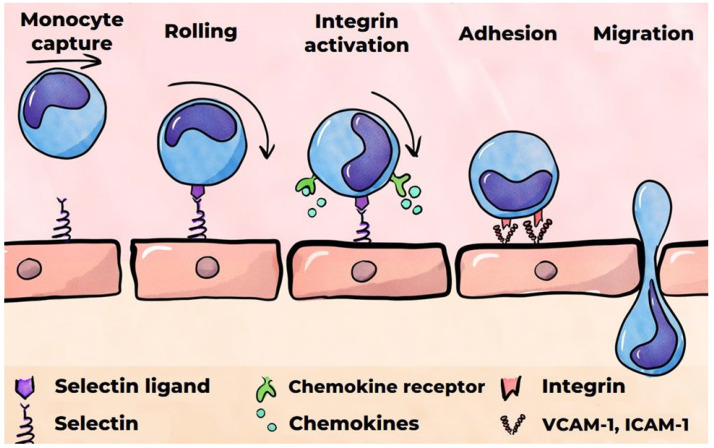
A monocyte’s migration into the subendothelial matrix.

**Figure 5 ijms-26-10962-f005:**
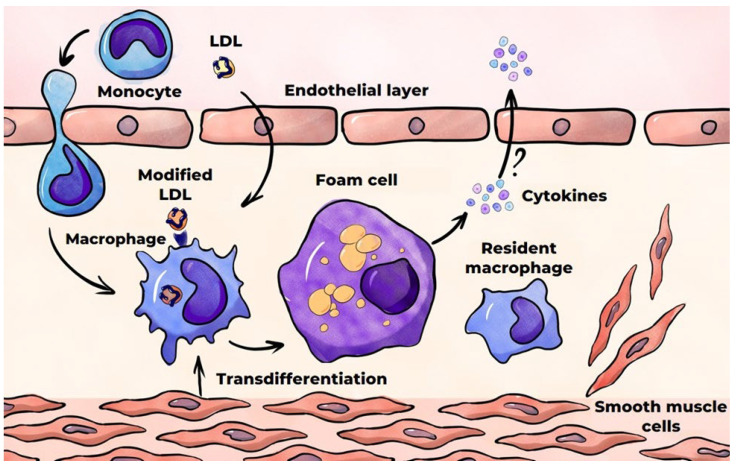
Foam cell formation in the course of atherosclerosis.

**Figure 6 ijms-26-10962-f006:**
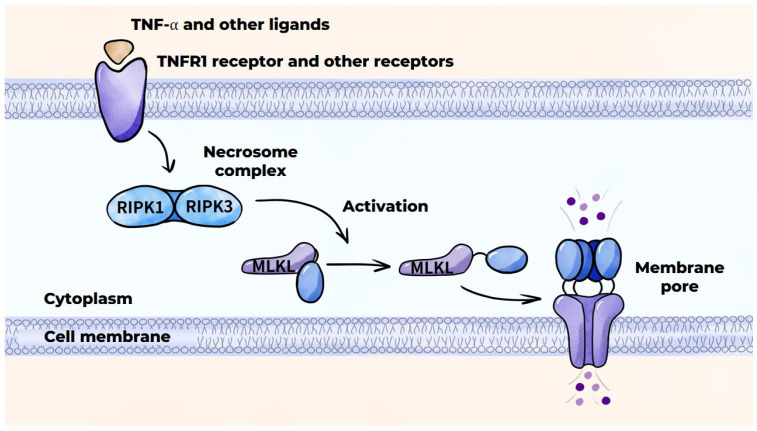
Programmed cell death mechanisms: necroptosis.

**Figure 7 ijms-26-10962-f007:**
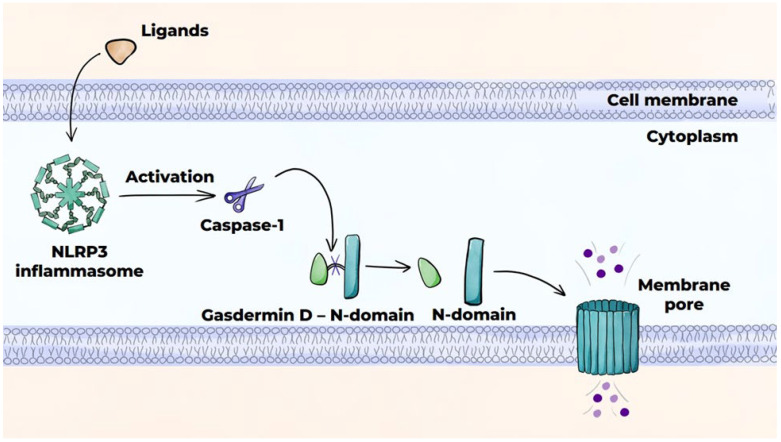
Programmed cell death mechanisms: pyroptosis (canonical pathway).

**Figure 8 ijms-26-10962-f008:**
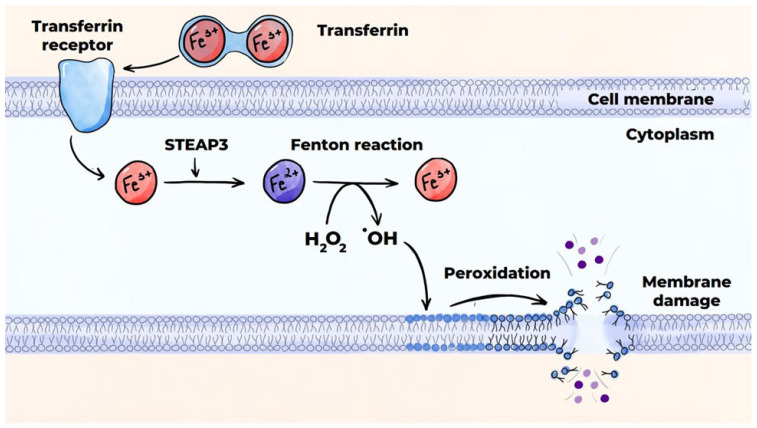
Mechanisms of programmed cell death: ferroptosis (non-canonical pathway). SLC11A2 is a divalent metal ion transporter protein belonging to the solute carrier family 11, member 2.

## Data Availability

No new data were created or analyzed in this study. Data sharing is not applicable to this article.

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
