# Peer review of "Monocytes/Macrophages and Atherogenesis"

_ijms, 2025, doi:10.3390/ijms262210962_

Round 1

Reviewer 1 Report

Comments and Suggestions for Authors

This article is a high-quality and authoritative review whose core strength lies in its systematic and comprehensive elaboration of the role of monocytes/macrophages in atherosclerosis. The authors not only clearly outline key biological processes—from the classification of monocyte subsets, the origins of tissue-resident versus monocyte-derived macrophages, to M1/M2 polarization, foam cell formation, and cell death—but also seamlessly integrate these cellular behaviors into the complete pathological sequence of atherogenesis, progression, and complications. The review is logically structured, transitioning smoothly from basic cellular functions to complex disease mechanisms and cutting-edge clinical translation, supported by key clinical trials such as CANTOS and COLCOT, making the discussion both in-depth and authoritative. Additionally, well-designed schematic diagrams effectively enhance the comprehension of complex concepts. Thus, this work provides readers with an invaluable and holistic knowledge framework in the field. However, several areas could be improved:

  1. The sections on inflammatory pathways and cell death mechanisms are somewhat repetitive. It is recommended to consolidate or streamline the "cell death mechanisms" part (e.g., apoptosis, necroptosis, pyroptosis, ferroptosis) to avoid redundancy and emphasize their specificity in atherosclerosis.

  1. The connection between mechanisms and therapeutics is not sufficiently tight. Although treatment strategies are mentioned, there is a lack of in-depth discussion on how existing drugs directly interact with monocyte/macrophage mechanisms. It is suggested to strengthen the link between the "treatment strategies" section and the preceding mechanisms—for example, how anti-IL-6 agents (e.g., Ziltivekimab and Canakinumab) affect macrophage function, or whether colchicine influences monocyte migration.

  1. While the existing figures are clear, they could be enhanced to better illustrate the dynamic nature of pathophysiological processes. For instance, adding a timeline diagram of "atherosclerotic plaque evolution and macrophage behavior" or including a schematic on "monocyte recruitment and plaque stability" would be beneficial.

  1. Some conclusions lack critical analysis. For example, the practical significance of M1/M2 polarization in humans and the discrepancies between animal models and human studies warrant further discussion. Additionally, it may be useful to briefly mention other potential sources of foam cells (e.g., endothelial cell transdifferentiation), beyond macrophages and smooth muscle cells.

Author Response

Response to Reviewer 1

We express our deep gratitude to Reviewer 1 for reviewing this manuscript and suggesting valuable improvements.

Point 1: The sections on inflammatory pathways and cell death mechanisms are somewhat repetitive. It is recommended to consolidate or streamline the "cell death mechanisms" part (e.g., apoptosis, necroptosis, pyroptosis, ferroptosis) to avoid redundancy and emphasize their specificity in atherosclerosis.

Response 1. To achieve better clarity, we introduced new subsections. We expect that this will improve the logical structure of the manuscript. In particular, we separated discussion of cell death mechanisms into the following subsections:

  • Page 15, line 628: ‘4.2. Macrophage death mechanisms in atherosclerosis: apoptosis
  • Page 16, line 672: ‘4.3. Macrophage death mechanisms in atherosclerosis: necroptosis
  • Page 16, line 687: ‘3.4.5. ‘Macrophage death mechanisms in atherosclerosis: pyroptosis
  • Page 17, line 699: ‘3.4.5. ‘Macrophage death mechanisms in atherosclerosis: ferroptosis

We highlighted that these cell death mechanisms are important for atherosclerotic plaque development and destabilization:

  • Page 15, lines 626-627 ‘A brief discussion of cell death mechanisms essential for atherosclerotic plaque development and destabilization is provided below.

Point 2. The connection between mechanisms and therapeutics is not sufficiently tight. Although treatment strategies are mentioned, there is a lack of in-depth discussion on how existing drugs directly interact with monocyte/macrophage mechanisms. It is suggested to strengthen the link between the "treatment strategies" section and the preceding mechanisms—for example, how anti-IL-6 agents (e.g., Ziltivekimab and Canakinumab) affect macrophage function, or whether colchicine influences monocyte migration.

Response 2. We introduced the following paragraph:

  • Pages 20-21, lines 859-866: ‘Anti- IL-1β and anti-IL-6 agents target downstream links of inflammatory pathways and do not directly target monocytes/macrophages. Nonetheless, there is recent evidence that colchicine can affect monocytes/macrophages. An ex vivo experiment demonstrated that colchicine reduced the proliferative activity of macrophages within human atherosclerotic plaques [196]. It also downregulated CD36 expression, decreasing macrophage lipid uptake and macrophage-to-foam cell transformation [197]. Colchicine also impeded circulating monocyte recruitment and adhesion molecule expression on monocytes in a murine model [198]

Point 3. While the existing figures are clear, they could be enhanced to better illustrate the dynamic nature of pathophysiological processes. For instance, adding a timeline diagram of "atherosclerotic plaque evolution and macrophage behavior" or including a schematic on "monocyte recruitment and plaque stability" would be beneficial.

Response 3. We introduced new figure 1 (page 2, line 49) that illustrates atherosclerotic plaque development.

Point 4. Some conclusions lack critical analysis. For example, the practical significance of M1/M2 polarization in humans and the discrepancies between animal models and human studies warrant further discussion. Additionally, it may be useful to briefly mention other potential sources of foam cells (e.g., endothelial cell transdifferentiation), beyond macrophages and smooth muscle cells.

Response 4. To our knowledge, there are no clinical trials studying agents that directly target M1/M2 macrophage polarization in humans. Nonetheless, potential therapies are under development. We introduced the following:

  • Page 21, lines 868-870: ‘New approaches, such as the use of macrophage-targeted nanoparticles or proteolysis-targeting chimera (PROTAC) technology, has been designed to target M1-to-M2 macrophage polarization [199]’
  • Page 21, lines 872-877: ‘Pioglitazone, a common anti-diabetic medication, was shown to regulate macrophage polarity towards the M2 phenotype in murine models [201,202]. Another anti-diabetic drug, gliclazide, reduced atherosclerotic plaque burden in a rabbit model. Gliclazide treatment was also associated with anti-inflammatory macrophage polarization, which was ascribed to the anti-NLRP3 inflammasome effect of this drug [203]

Considering other sources of foam cells, we introduced the following:

  • Page 14, lines 599-602: ‘The aspects of macrophage and SMCs interaction in foam cell formation are comprehensively outlined in the review by Xiang et al. [136]. Other sources of foam cells may include stem progenitor cells and endothelial cells, although little is known about the extent of their contribution to the resulting foam cell pool [137,138].

Point 5. The English could be improved to more clearly express the research.

Response 5. The manuscript was edited by the Academic English editing team of MDPI.

Reviewer 2 Report

Comments and Suggestions for Authors

Monocytes/macrophages are known to be key players in atherogenesis, driving the disease by entering the artery wall, differentiating into macrophages, and becoming foam cells through lipid uptake. They orchestrate the inflammatory response, produce pro-inflammatory cytokines like TNF-α, and contribute to plaque development and instability. Different subsets of monocytes and macrophages play roles in promoting or resolving the disease. This is the rationale for this review.
The review cited 183 references.
The abstract reflects the article's content but resembles a collection of obvious connotations. It would be better if the authors emphasized the uniqueness of this review and how it differs from similar ones. There are many reviews on this topic, many with similar titles, so it's important to understand what makes this particular review unique. It's also important to indicate which databases were searched and the period covered.
The review structure is as follows:
1.    Introduction
The introduction is logical and brief, and at the end, the objective is stated—an overview of current data on the classification and functions of monocytes/macrophages.
2.    General characteristics of monocytes/macrophages
2.1. Monocytes
This section presents basic information about monocytes, their subpopulations, origins, and surface markers. This section includes a figure:
Figure 1. Human monocyte subpopulations.
Classical, non-classical, and intermediate forms are presented. The figure is clear, like in a textbook.
2.2. Macrophages
This section presents basic information about macrophages, their origins, characteristics, and differentiation pathways.
Figure 2. M1/M2 macrophage polarization. The drawing is obvious, like in a textbook.
2.3. Dendritic cells.
A minimal amount of information about these cells is presented.
3.    Participation of monocytes/macrophages in atherogenesis
3.1. Macrophage uptake of modified LDLs in the arterial wall
It would be better if LDL were deciphered. The mechanisms of LDL uptake by cells are briefly described.
3.2. Inflammatory activation of macrophages and their interaction with the endothelium
The mechanisms of inflammatory activation of macrophages are described, at the level of surface molecules and transcription factors. Monocyte migration into the subendothelial matrix is also discussed.
Figure 3. Monocyte migration into the subendothelial matrix
3.3. Formation of foam cells
The mechanisms of foam cell formation are described. Figure 4. Foam cell formation in the course of atherosclerosis.
3.4. Formation of atherosclerotic plaque
The mechanisms of atherosclerotic plaque formation are described, at different stages, and using a variety of cell death mechanisms, including apoptosis, necroptosis, pyroptosis, and ferroptosis.
3.5. Macrophages and the local immune response
This section describes the involvement of other immune system cells in the development of atherosclerosis—T cells, Tregs, and B cells.
3.6. Macrophages and Resolution of Inflammation
The difficulties of macrophage migration in atherosclerosis and the contribution of this phenomenon to the development of inflammation are described.
4.    Monocyte/macrophage-directed therapeutics in atherosclerosis
A cross-section of clinical trials and research studies focusing on the effects of monocytes/macrophages in atherosclerosis is presented.
5.    Conclusion
A logical and brief conclusion on the role of monocytes/macrophages in atherosclerosis.
Thus, the report appears very trivial, more like a fragment from a textbook. The advantages of the review are its logicality and brevity, as well as good illustrations. The disadvantage of the review is its obviousness—it's unclear what makes it unique and necessary.
For example, we were able to find several reviews on a similar topic that are not cited in this review.
Amengual J, Barrett TJ. Monocytes and macrophages in atherogenesis. Curr Opin Lipidol. 2019 Oct;30(5):401-408. doi: 10.1097/MOL.0000000000000634.
Farahi L, Sinha SK and Lusis AJ (2021) Roles of Macrophages in Atherogenesis. Front. Pharmacol. 12:785220. doi: 10.3389/fphar.2021.785220
Hou, P., Fang, J., Liu, Z. et al. Macrophage polarization and metabolism in atherosclerosis. Cell Death Dis 14, 691 (2023). https://doi.org/10.1038/s41419-023-06206-z
The review needs to be improved by emphasizing what makes it unique, perhaps by adding a unique section that is not covered in similar publications.

Summary: While the review "Monocytes/macrophages and atherogenesis" is well written and illustrated, it could be significantly improved. First and foremost, it should be supplemented with references to similar reviews and a clear statement of the uniqueness and novelty of this review. Currently, the review presents obvious and well-known facts. 
It is not obvious to the reviewer what gap in the field of modern biomedicine this review fills.
To avoid a textbook format, it is necessary to highlight what has become known in recent years. The best solution would be to create a new, unique section within this review—for example, one devoted to cellular metabolism, lipid profiles, or something else. A section indicating the databases and period covered by the search should also be created; this information should be included in the abstract as well. The section on "dendritic cells" needs to be expanded.

Author Response

Response to Reviewer 2

We sincerely thank Reviewer 2 for thorough evaluation of this manuscript and valuable suggestions.

Point 1. The abstract reflects the article's content but resembles a collection of obvious connotations. It would be better if the authors emphasized the uniqueness of this review and how it differs from similar ones. There are many reviews on this topic, many with similar titles, so it's important to understand what makes this particular review unique. It's also important to indicate which databases were searched and the period covered.

Response 2. We added the following at the end of the abstract:

  • Page 1, lines 28-31: ‘…and highlights existing gaps in evidence. This review is primarily aimed at readers with a background in clinical medicine who are interested in the involvement of monocytes/macrophages in atherogenesis’.

To indicate databases that were searched and the period that was covered, the following was introduced:

  • Page 2, lines 63-69: ‘In preparation for this paper, PubMed, Medline, and Web of Science databases were searched for publications. The search encompassed animal and human studies, ranging from basic research to randomized clinical trials, meta-analyses, and reviews. Original research articles with consistent methodology published from 2010 to the present were given priority, although some earlier articles providing crucial data were also discussed and cited. State-of-the-art review papers were also cited to refer the reader to additional in-depth resources on specific topics

Point 2. 2.3. Dendritic cells. A minimal amount of information about these cells is presented.

Response 2. We introduced the following:

  • Page 10, lines 433-438: ‘Monocyte-derived dendritic cells can differentiate into two distinct subtypes: (1) dendritic cells that promote inflammation and immune cell activation by presenting antigens to T cells and secreting pro-inflammatory cytokines [87] and (2) dendritic cells that promote immune tolerance by presenting antigens to T reg cells and secreting anti-inflammatory cytokines [88]. Dendritic cells’ role in atherogenesis is explored in detail elsewhere [89,90]’.

Point 3. 3.1. Macrophage uptake of modified LDLs in the arterial wall. It would be better if LDL were deciphered.

Response 3. LDLs were deciphered in the heading (page 10, line 440) and at the first mention in the text (page 10, line 446).

Point 4. Thus, the report appears very trivial, more like a fragment from a textbook. The advantages of the review are its logicality and brevity, as well as good illustrations. The disadvantage of the review is its obviousness—it's unclear what makes it unique and necessary.
For example, we were able to find several reviews on a similar topic that are not cited in this review.
Amengual J, Barrett TJ. Monocytes and macrophages in atherogenesis. Curr Opin Lipidol. 2019 Oct;30(5):401-408. doi: 10.1097/MOL.0000000000000634.

Farahi L, Sinha SK and Lusis AJ (2021) Roles of Macrophages in Atherogenesis. Front. Pharmacol. 12:785220. doi: 10.3389/fphar.2021.785220

Hou, P., Fang, J., Liu, Z. et al. Macrophage polarization and metabolism in atherosclerosis. Cell Death Dis 14, 691 (2023). https://doi.org/10.1038/s41419-023-06206-z

The review needs to be improved by emphasizing what makes it unique, perhaps by adding a unique section that is not covered in similar publications.

Response 4. While there are a number of other high-quality reviews on the same topic, we believe that this particular manuscript has its own advantages. One of the main advantages of this manuscript is that it reviews a number of resent original research papers. And in doing so, it states what data were obtained in experimental settings, and what was only hypothesized; what parts of knowledge about monocyte/macrophages are still incomplete, and what research results are contradictory or inconclusive. Furthermore, the manuscript clearly indicates existing gaps in evidence. We believe that this is important for further research.

We also added the following:

  • Page 18, lines 734-761: ‘4.7. Fibroblasts in the atherosclerotic plaque. The endothelial-mesenchymal transition and contractile-to-synthetic phenotype switching of SMCs are considered the main sources of cells with the synthetic phenotype in the atherosclerotic plaque [158]. Nonetheless, recent findings indicate that fibroblasts may also be involved in atherogenesis from its earliest stages [159]. Fibroblasts are heterogeneous cells of mesenchymal origin, located in adventitia of healthy vessels, where they reside in a quiescent state [160]. In response to numerous pro-inflammatory stimuli, such as TGF-β, IL-10, platelet-derived growth factor (PDGF), vascular endothelial growth factor (VEGF), epidermal growth factor receptor (EGFR), and amphiregulin (AREG) released by macrophages, fibroblasts become activated [161]. It was postulated that neovascularization, observed in atherosclerotic lesions, can facilitate the delivery of fibroblasts into the plaque [160]. Nonetheless, it is not clear to which extent adventitious fibroblasts contribute to the pool of cells with the synthetic phenotype in atherosclerotic lesions. The high heterogenicity and plasticity of fibroblasts, as well as the ability of other types of cells to transform into fibroblast-like cells, hinder the clear distinction of fibroblasts and fibroblast-like cell subsets in the atherosclerotic plaque. In addition, macrophages can potentially differentiate into fibroblast-like cells and produce components of extracellular matrix, including collagen fibers [162]. The macrophage-to-myofibroblast transition is a phenomenon observed in fibrotic diseases [163]. It is not clear whether this transition occurs in atherosclerosis.

In addition to components of extracellular matrix, activated fibroblasts produce pro-inflammatory cytokines, including IL-1, TGF-β, IL-6, reactive oxygen species, and growth factors, such as VEGF, maintaining a proinflammatory microenvironment in the atherosclerotic lesions [164]. Different subsets of activated fibroblasts produce a number of MMPs and tissue inhibitors of MMPs (TIMPs), contributing to atherosclerotic plaque stabilization and destabilization [164].

It was postulated that fibroblasts play an important role in plaque calcification [165] and neointimal growth after coronary artery stenting [166]’.

Moreover, section #4 ‘Monocyte/macrophage-directed therapeutics in atherosclerosis’ was expanded. We added the following:

  • Pages 20-21, lines 859-866: ‘Anti- IL-1β and anti-IL-6 agents target downstream links of inflammatory pathways and do not directly target monocytes/macrophages. Nonetheless, there is recent evidence that colchicine can affect monocytes/macrophages. An ex vivo experiment demonstrated that colchicine reduced the proliferative activity of macrophages within human atherosclerotic plaques [196]. It also downregulated CD36 expression, decreasing macrophage lipid uptake and macrophage-to-foam cell transformation [197]. Colchicine also impeded circulating monocyte recruitment and adhesion molecule expression on monocytes in a murine model [198]
  • Page 21, lines 868-870: ‘New approaches, such as the use of macrophage-targeted nanoparticles or proteolysis-targeting chimera (PROTAC) technology, has been designed to target M1-to-M2 macrophage polarization [199]’
  • Page 21, lines 872-877: ‘Pioglitazone, a common anti-diabetic medication, was shown to regulate macrophage polarity towards the M2 phenotype in murine models [201,202]. Another anti-diabetic drug, gliclazide, reduced atherosclerotic plaque burden in a rabbit model. Gliclazide treatment was also associated with anti-inflammatory macrophage polarization, which was ascribed to the anti-NLRP3 inflammasome effect of this drug [203]

The reviews you kindly provided were cited (page 2, line 59; page 2, line 62; page 8, line 328).

Round 2

Reviewer 2 Report

Comments and Suggestions for Authors

The article was received for re-review after revision. The authors responded to all questions and took into account the reviewer's comments.
Specifically, the abstract was supplemented, the search terms for this review were specified, and the article selection criteria were outlined. The "dendritic cells" section was expanded.
The authors kindly agreed to include the suggested references to similar reviews in their text. The authors outlined the unique features of this review. Specifically, the strengths of the review include the discussion of many recently published studies, clarifying which data have experimental confirmation and which are merely hypothetical. The authors expanded their review with new information on fibroblasts in atherosclerotic plaque. Importantly, Section 4, which is quite interesting and unique, was also expanded.

The reviewer thanks the authors for their attentive response to questions and comments. This version of the article may be published.